# Spatial patterns and determinants of low utilization of delivery care service and postnatal check-up within 2 months following birth in Ethiopia: Bivariate analysis

**Shegaw Mamaru Awoke**[1]*, **Frezer Tilahun Getaneh**[2], **Muluwerk Ayele Derebe**[3]

**1** Department of Statistics, Assosa University, Assosa, Ethiopia, **2** Department of Statistics, Mekdela Amba University, Mekdela Amba, Ethiopia, **3** Department of Statistics, Bahir Dar University, Bahir Dar, Ethiopia

* mamarushegaw@gmail.com

## Abstract

### Background

Home delivery is a nonclinical childbirth practice that takes place in one's home with or without traditional birth attendants and postnatal care is the care given to the mother and her newborn baby; according to world health organization (WHO), the postnatal phase, begins one hour after birth and lasts six weeks (42 days). This paper aimed to study the spatial pattern and determinant factors of low utilization of delivery care (DC) services and postnatal check-up (PNC) after live births in Ethiopia.

### Methods

This study used the 2016 Ethiopian Demographic and Health Survey data as a source. A total weighted samples of 11023 women-children pairs were included. The bivariate binary logistic regression analyses with spatial effect were modeled using SAS version 9.4 and ArcGIS version 10.8 was used for mapping.

### Results

The spatial distribution of low utilization of delivery care service and postnatal check-up were significantly clustered in Ethiopia (Moran's I statistic 0.378, P-value < 0.001 and 0.177, P-value < 0.001 respectively). Among 11023 children-women pair, the prevalence of home delivery and no postnatal check-up within two months following birth were 72.6% and 91.4% respectively. The Liben, Borena, Guji, Bale, Dolo and Zone 2 were predicted to have high prevalence of home delivery and part of Afder, Shabelle, Korahe, Dolo and Zone 2 were high risk areas of no postnatal checkup.

### Conclusion and recommendations

Lack of occupation, region, large family size, higher birth order, low utilization of antenatal care visit, unable to access mass media, big problem of health facility distance and the spatial variable were found to be jointly significant predictors of low utilization of DC and PNC in

program. The Ethiopian Demographic and Health Survey, as part of the Demographic and Health Survey (DHS) program, offers publicly available data. Researchers can access the data by becoming authorized users and registering on the website http://www.dhsprogram.com. Once access permission is granted, users can download the datasets from the required countries without any cost. This ensures that all the data used in the survey's findings are freely accessible to researchers. URL: https://dhsprogram.com/data/dataset/Ethiopia_Standard-DHS_2016.cfm?flag=1.

**Funding:** The author(s) received no specific funding for this work.

**Competing interests:** The authors have declared that no competing interests exist.

**Abbreviations:** DC, Delivery care; EAs, Enumeration areas; EDHS, Ethiopian demographic and health; PNC, Postnatal check-up.

Ethiopia. Whereas older age, being reside in rural area and low wealth status affects delivery care service utilization. We suggest health providers, policy makers and stakeholders consider those variables with priority given to Liben, Borena, Guji, Bale, Dolo, Zone 2, Afder, Shabelle and Korahe, where home delivery and no PNC were predicted relatively high. We also recommend researchers to conduct further studies using latest survey data set.

## Introduction

Home delivery is a nonclinical childbirth practice that takes place in one's home with or without traditional birth attendants [1]. Maternal and child mortality due to obstetric complications; including hemorrhage, hypertension, sepsis, abortion, and embolism is a critical issue specially in developing nations [2]. Access to health facilities and skilled personnel is crucial for safe motherhood, reducing maternal and newborn mortality by increasing women's utilization to emergency obstetric care [3].

Evidence demonstrated that unskilled delivery practices were directly responsible for around 66% of all maternal deaths worldwide, and more than 50% in developing countries [4,5]. To promote mother and child health, institutional delivery services have been recommended [6,7]. Despite the fact that skilled health workers are known to be crucial actors in reducing maternal mortality by preventing and managing difficulties during pregnancy and childbirth, a number of women have died as a result of giving birth without the presence of a skilled health worker [8–10].

Postnatal care (PNC) also known as postpartum is the care given to the mother and her newborn baby; according to WHO, the postnatal phase, begins one hour after the placenta is delivered and lasts six weeks (42 days) [11]. For both mothers and their newborn child to survive, the postnatal period is critical. Missing out on postnatal care during this critical time could lead to complications and the deaths of both the mother and the baby [12]. As a result, the World Health Organization (WHO) recommends that mothers and newborns should receive PNC at health facilities for at least 24 hours following birth, if the birth occurs in a health facility. If the birth takes place at home, the first postnatal contact should take place as soon as feasible after the birth, ideally within 24 hours. A minimum of three further postnatal health assessments for both women and newborns are recommended on day 3 (48–72 hours), within 7–14 days, and at 6 weeks after birth [13].

Maternal health care is essential for the prevention and management of physical and mental impairments, but is neglected in developing countries; about half of maternal deaths occur within the first week following birth, with the majority occurring within the first 24 hours due to women and their newborns' lack of access to treatment during the early postnatal period [14]. In 2015, the global maternal mortality ratio (MMR) was 216 maternal deaths per 100,000 live births during pregnancy and delivery [15]. The majority of maternal deaths (99%) occur from preventable causes in developing countries, including Ethiopia [16].

Postnatal care research in developing countries is scarce in comparison to antenatal care and skilled attendance at birth [12]. Few recent findings indicated that numerous characteristics, including Marital status, knowledge of the mothers on PNC service, place of residence, media exposure, place of delivery, residency, ANC follow-up, and administrative region, have been considered as determinants of postnatal care service utilization [17,18]. However, several studies have mainly neglected the potential impacts of geographical factors [19,20], and parallel effects (bivariate effects) of the place of delivery on the evaluation of PNC [21].

Even if studies on PNC and DC utilization and possible determinants were conducted separately [19,21], maternal and child mortality can be reduced by acquiring PNC and DC jointly. Thus, we are motivated to conduct a study on the geographical pattern and determinant factors of low utilization of delivery care services and postnatal check-up following live births in Ethiopia.

## Methods

### Study area

This study was conducted in Ethiopia, which is located in the horn of Africa and has nine regional states namely Tigray, Afar, Amhara, Oromia, Somali, Benishangul Gumuz, Southern Nations, Nationalities and People's Region (SNNPR), Gambella, and Harari; two administrative cities namely Addis Ababa and Dire Dawa and more than 72 administrative zones [22].

### Data source and study population

The Demographic and Health Survey (DHS) data is publicly available and can be accessed by researchers through the DHS Program's website at www.dhsprogram.com. Researchers can register on the website and request access to the datasets they need. Once access is granted, researchers can download the datasets free of charge. Accordingly, the 2016 Ethiopian Demographic and Health Survey data was used for the current study. The source population was all reproductive age group (15–49 years) women in Ethiopia in EDHS 2016. The study population was women who gave at least one live birth in the last 5 years preceding the survey for the most recent birth.

### Sample size and sampling procedure

After the data has been cleaned, a total of 11023 women-children pairs were included in this study. Participants were selected based on a stratified two-stage cluster sampling technique. Weighted values were used to increase the representativeness of the sample data. Sample weights were calculated in each child's record (KR) EDHS datasets. After excluding clusters with zero coordinates a total of 643 clusters were included.

### Ethics approval and consent to participate

Permission to access the data for this study was obtained from the major demographic and health survey (DHS) program through reasonable online request via www.dhsprogram.com. The data used in this study were available without individual identifiers. The institutional review board approved procedures for DHS public-use datasets do not allow specific households or sample clusters to be identified. The geographic identifiers are available only for the enumeration areas (EAs) as a whole, not for particular household addresses. All methods were carried out in accordance with relevant guidelines and regulations. As the data was obtained from a secondary source informed consent was not required by the ethical review board.

### Study variables

### Outcome variables

Two binary outcome variables were considered in this study. These are place of delivery (home, coded as "1" or health facility, coded as "0") and postnatal check-up utilization within two months following birth (No, coded as "1" or Yes, coded as "0").

### Independent variables

From the 2016 EDHS data set, socio-demographic and obstetric characteristics that are associated with delivery care (DC) and postnatal care (PNC) were taken as independent variables. This include mother age, mother education level, mother occupation, husband/partner education level, husband/partner occupation, region, residence, religion, sex of household head, family size, current marital status, access to mass media, preceding birth interval, birth order, antenatal care visit, household wealth index, distance to health facility and health insurance.

### Data management and analysis

The data was extracted and managed using SPSS version 26 and STATA version 17 software. Sample weighting was done before further analysis. Priori of model fitting, Chi-square test of association was carried out for the data to examine the relationship between the two response variables (DC and PNC) and each independent variables. Data were then analyzed using SAS version 9.4 and ArcGIS version 10.8 was used for mapping.

### Spatial analysis

**Spatial autocorrelation.**   In order to determine whether home deliveries and lack of postnatal check-ups were scattered, clustered, or randomly distributed in Ethiopia, spatial autocorrelation (Global Moran's I statistic) was used. A Moran's I values close to − 1 indicates that home delivery or no postnatal checkup was dispersed, close to + 1 indicates clustered, and if Moran's I value close to zero indicates randomly distributed [21].

$$\text{Moran's } I = \frac{n \sum_i^n \sum_j^n wij(yi - \bar{y})(yj - \bar{y})}{\left(\sum_i^n \sum_j^n wij\right) \sum_i (yi - \bar{y})^2} \tag{1}$$

Where: $y_i$ represents the vector of observations at n different locations, and $w_{ij}$ are elements of a spatial weight matrix.

**Spatial weight matrix.**   The majority of spatial models determine whether one location is the spatial neighbor of another location. A spatial weight matrix is a square symmetric matrix of dimension $n{\times}n$ with the $(i,j)^{th}$ element equal to 1 if regions $i$ and $j$ are neighbors, and zero otherwise [23]. Consider a spatial weighted matrix $W$ of size $n{\times}n$, given by:

$$W = \begin{bmatrix} 0 & w_{12} & w_{13} & \cdots & w_{1N} \\ w_{21} & 0 & w_{23} & \ldots & w_{2N} \\ w_{31} & w_{32} & 0 & \ldots & w_{3N} \\ \vdots & \vdots & \vdots & \ddots & \vdots \\ w_{N1} & w_{N2} & w_{N3} & \cdots & 0 \end{bmatrix} \tag{2}$$

$$\text{Where, } W_{ij} = \begin{cases} 1, & \text{if two areas are adjacent} \\ 0, & \text{other wise} \end{cases}$$

Assuming the weights $W_{ij}$ are binary, they simply identify which elements of the computation are to be included or excluded in the calculation.

**Hot spot analysis.**   Hot spot analysis identifies statistically significant clustering areas using latitude and longitude coordinate readings that were taken at the nearest community center for clusters. The Z score and p value from the Getis-Ord Gi statistic shows where the high and low values cluster spatially. The hot spot region is created when high values in the

given data are surrounded by other high values, and the cold spot area is created when low values are surrounded by other low values [24].

**Spatial interpolation.** Spatial interpolation is a procedure of estimating the values of properties at un-sampled locations based on the set of observed values at known locations [3]. A large number of interpolation methods have been developed for use with point, line, and area data. No matter which interpolation technique is used, the derived values are only estimates of what the real values should be at a particular location [25]. Geo-statistical interpolation techniques (kriging) utilize the statistical properties of the measured points and were used in the present study.

**Bivariate binary logistic regression.** It is natural to measure two binary responses in various applications and wish to model them jointly as a function of some covariates. A natural regression model for such data is then created by combining two logistic regressions with an equation for the odds ratio. The odds ratio is a logical way to assess the relationship between two binary variables because the responses are frequently correlated. The bivariate odds ratio model, commonly referred to as the bivariate logistic regression model. Let $Y_1$ (DC) and $Y_2$ (PNC) be the two response variables, then their joint and marginal probability can be presented by (Table 1) [26].

Based on Table 1 the joint probability function follows a multinomial distribution defined by:

$$P(Y_{11} = y_{11}, \; Y_{10} = y_{10}, \; Y_{01} = y_{01}, \; Y_{00} = y_{00}) = \prod_{g=0}^{1} \prod_{h=0}^{1} \frac{p_{gh}^{y_{gh}}}{y_{gh}!}, \; 0 < p_{gh} < 1 \qquad (3)$$

Where: $g, h = 0.1$; $y_{gh} = 0, 1$ and $p_{00} = 1 - p_{11} - p_{10} - p_{01}$

Furthermore, the bivariate logistic regression (BLR) model can be written as follows:

$$\begin{aligned} g_1(x) &= \text{logit}(p_1(x)) = \beta_1^T x \\ g_2(x) &= \log it(p_2(x)) = \beta_2^T x \\ g_3(x) &= \ln \ln \psi_1(x) = \beta_3^T x \end{aligned} \qquad (4)$$

Where, $x = \begin{bmatrix} 1 & x_1 & x_2 & \cdots & x_k \end{bmatrix}^T$ is a vector of the covariate, $\beta_1^T = [\beta_{01} \; \beta_{11} \; \beta_{21} \cdots \beta_{k1}]$, $\beta_2^T = [\beta_{02} \; \beta_{12} \; \beta_{22} \cdots \beta_{k2}]$, and $\beta_3^T = [\beta_{03} \; \beta_{13} \; \beta_{23} \cdots \beta_{k3}]$ are the parameters, $p_1(x)$ is the marginal probability function of $Y_1$, $p_2(x)$ is the marginal probability function of $Y_2$, and $\psi_1(x)$ is the odds ratio that shows an association between $Y_1$ and $Y_2$.

According to [27], the joint probability of $p_{11}(x)$ can be obtained as follows:

$$p_{11}(x) = \begin{cases} \dfrac{a_1 - \sqrt{a_1^2 + b_1}}{2(\psi_1(x) - 1)}, \; \psi_1(x) \neq 1 \\[3ex] p_1(x)p_2(x), \; \psi_1(x) = 1 \end{cases} \qquad (5)$$

Where: $a_1 = 1 + (\psi_1(x) - 1)(p_1(x) + p_2(x))$, $b_1 = -4\psi_1(x)(\psi_1(x) - 1)p_1(x)p_2(x)$ with $p_1(x)$ and $p_2(x)$ are the marginal probabilities. Furthermore, based on Table 1, the joint

**Table 1. Joint and marginal probability response variables (DC and PNC).**

| DC ($Y_1$) | PNC ($Y_2$) | | Total |
|---|---|---|---|
| | $Y_2 = 1$ | $Y_2 = 0$ | |
| $Y_1 = 1$ | $p_{11}$ | $p_{10}$ | $p_{1+}$ |
| $Y_1 = 0$ | $p_{01}$ | $p_{00}$ | $p_{0+} = 1 - p_{1+}$ |
| Total | $p_{+1}$ | $p_{+0} = 1 - p_{+1}$ | $p_{++} = 1$ |

probabilities of $p_{10}(x)$, $p_{01}(x)$, and $p_{00}(x)$ are obtained as:

$$\begin{aligned}
p_{10}(x) &= p_1(x) - p_{11}(x)\\
p_{01}(x) &= p_2(x) - p_{11}(x)\\
p_{00}(x) &= 1 - p_{11}(x) - p_{10}(x) - p_{01}(x)\\
&= 1 - p_1(x) - p_2(x) + p_{11}(x)
\end{aligned} \tag{6}$$

The marginal probability function and the odds ratio are defined by [28]:

$$\begin{aligned}
p_1(x) &= p_{1+}(x) = \frac{e^{\beta_1^T x}}{1 + e^{\beta_1^T x}},\\
p_2(x) &= p_{+1}(x) = \frac{e^{\beta_2^T x}}{1 + e^{\beta_2^T x}},\\
\psi_1(x) &= \frac{p_{11}(x)p_{00}(x)}{p_{10}(x)p_{01}(x)},
\end{aligned} \tag{7}$$

**Bivariate binary with spatial effect model.** The bivariate binary with spatial effect model is an extension of the ordinary bivariate binary logistic regression model that incorporates a new predictor, which is a weighted average of the proportion of the response variables (DC and PNC) for the neighboring sites (zones) in to the models explanatory variables. This new predictor is known as auto-covariance, thus the usual bivariate binary logistic regression model is changed into as follows [28,29]:

$$\begin{aligned}
g_1(x) &= \text{logit}(p_1(x)) = \beta_1^T x + \rho Si\\
g_2(x) &= \text{logit}(p_2(x)) = \beta_2^T x + \rho Si\\
g_3(x) &= \ln \ln \psi_1(x) = \beta_3^T x + \rho Si
\end{aligned} \tag{8}$$

Where: $\rho$ is the coefficient of the auto-covariance variable ($S_i$) at any site (zone), which is calculated as [29,30]:

$$S_i = \frac{\sum_{j=1}^{k_i} w_{ij} y_j}{\sum_{j=1}^{k_i} w_{ij}} \tag{9}$$

Where: $y_j$ is the response value of $y$ at site $j$ among site i's set of $K_i$ neighbors and $w_{ij}$ is the $(i,j)^{th}$ element of the spatial weight matrix.

## Results

This study comprised a total weighted sample of 11023 women-child pair. The association between the predictor variables and the two response variables; delivery care (DC) and postnatal checkup (PNC) was revealed in (Table 2). Among newborns delivered at home, the majority, 2409 (30.1%) were born from mothers between the ages of 25–29 years. As both the education level of women and husband increases the prevalence of newborns delivered at home and not receive postnatal checkup decreases. High home delivery and no postnatal checkup were more experienced by newborns from a non-working mother; 4606 (57.6%) and 5662 (56.2%) respectively. Regarding to residence, a woman who lives in rural area used to deliver at home and not follow up postnatal checkup at a prevalence rate of 96% and 90.0%, respectively. Women who had antenatal care visit accounted for 52.4% of health facility delivery and 50.4% postnatal checkup utilization. Moreover, highest prevalence of home delivery

**Table 2. Association of socio-demographic and obstetric characteristics with DC and PNC; EDHS 2016.**

| Variables | Categories | Weighted freque (%) | Place of Delivery | | | Postnatal checkup | | |
|---|---|---|---|---|---|---|---|---|
| | | | Home (%) | HF (%) | $X^2$-p value | No (%) | Yes (%) | $X^2$-p value |
| Age | 15–19 | 378 (3.4) | 223 (2.8) | 155 (5.1) | 0.000 | 346(3.4) | 32 (3.4) | 0.136 |
| | 20–24 | 2067 (18.8) | 1339 (16.7) | 728 (24.1) | | 1892 (18.8) | 176 (18.6) | |
| | 25–29 | 3353 (30.4) | 2409 (30.1) | 944 (31.2) | | 3043 (30.2) | 310 (32.8) | |
| | 30–34 | 2490 (22.6) | 1889 (23.6) | 601 (19.9) | | 2288 (22.7) | 202 (21.4) | |
| | 35–39 | 1772 (16.1) | 1331 (16.6) | 441 (14.6) | | 1644 (16.3) | 128 (13.6) | |
| | 40–44 | 723 (6.6) | 599 (7.5) | 124 (4.1) | | 648 (6.4) | 75 (7.9) | |
| | 45–49 | 239 (2.2) | 207 (2.6) | 32 (1.1) | | 219 (2.2) | 21 (2.2) | |
| Mother's education | No educ. | 7284 (66.1) | 6042 (75.6) | 1242 (41.1) | 0.000 | 6760 (67.1) | 523 (55.5) | 0.000 |
| | Primary | 2950 (26.8) | 1819 (22.7) | 1131 (37.4) | | 2672 (26.5) | 279 (29.6) | |
| | Secondary | 513 (4.7) | 112 (1.4) | 401 (13.3) | | 423 (4.2) | 91 (9.7) | |
| | Higher | 274 (2.5) | 23 (0.3) | 251 (8.3) | | 225 (2.2) | 49 (5.2) | |
| Mother's Occupation | Not working | 6127 (55.6) | 4606 (57.6) | 1521 (50.3) | 0.000 | 5662 (56.2) | 465 (49.3) | 0.000 |
| | Working | 4896 (44.4) | 3391 (42.4) | 1505 (49.7) | | 4418 (43.8) | 478 (50.7) | |
| Husband education | No educ. | 5358 (48.6) | 4358 (54.5) | 1000 (33.0) | 0.000 | 4981 (49.4) | 376 (39.9) | 0.000 |
| | Primary | 4304 (39.0) | 3180 (39.8) | 1124 (37.1) | | 3925 (38.9) | 379 (40.2) | |
| | Secondary | 852 (7.7) | 345 (4.3) | 507 (16.8) | | 748 (7.4) | 104 (11.0) | |
| | Higher | 509 (4.6) | 114 (1.4) | 395 (13.1) | | 425 (4.2) | 84 (8.9) | |
| Husband occupation | Not working | 960 (8.7) | 747 (9.3) | 213 (7.0) | 0.000 | 894 (8.9) | 66 (7.0) | 0.000 |
| | Working | 10063 (91.3) | 7250 (90.7) | 2813 (93.0) | | 9186 (91.1) | 877 (93.0) | |
| Household head | Male | 9494 (86.1) | 6986 (87.4) | 2508 (82.9) | 0.000 | 8708 (86.4) | 786 (83.4) | 0.01 |
| | Female | 1529 (13.9) | 1011 (12.6) | 518 (17.1) | | 1372 (13.6) | 157 (16.6) | |
| Marital status | Unmarried | 561 (5.1) | 369 (4.6) | 192 (6.3) | 0.000 | 507 (5.0) | 53 (5.6) | 0.430 |
| | Married | 10463 (94.9) | 7629 (95.4) | 2834 (93.7) | | 9573 (95.0) | 890 (94.4) | |
| Religion | Orthodox | 3772 (34.2) | 2339 (29.2) | 1433 (47.4) | 0.000 | 3344 (33.2) | 428 (45.4) | 0.000 |
| | Catholic | 104 (0.9) | 83 (1.0) | 21 (0.7) | | 93 (0.9) | 10 (1.1) | |
| | Protestant | 2329 (21.1) | 1733 (21.7) | 596 (19.7) | | 2131 (21.1) | 198 (21.0) | |
| | Muslim | 4561 (41.4) | 3605 (45.1) | 956 (31.6) | | 4259 (42.3) | 302 (32.0) | |
| | Others | 257 (2.3) | 238 (3.0) | 19 (0.6) | | 253(2.5) | 5(0.5) | |
| Region | Tigray | 716 (6.5) | 294 (3.7) | 422 (14.0) | 0.000 | 597 (5.9) | 119 (12.6) | 0.000 |
| | Afar | 114 (1.0) | 97 (1.2) | 17 (0.6) | | 107 (1.1) | 7 (0.7) | |
| | Amhara | 2073 (18.8) | 1479 (18.5) | 594 (19.6) | | 1852 (18.4) | 220 (23.4) | |
| | Oromo | 4850 (44.0) | 3903 (48.8) | 947 (31.3) | | 4565 (45.3) | 286 (30.4) | |
| | Somalia | 508 (4.6) | 417 (5.2) | 91 (3.0) | | 478 (4.7) | 30 (3.2) | |
| | Benishangul | 122 (1.1) | 89 (1.1) | 33 (1.1) | | 105 (1.0) | 16 (1.7) | |
| | SNNP | 2296 (20.8) | 1664 (20.8) | 632 (20.9) | | 2095 (20.8) | 201 (21.3) | |
| | Gambela | 26 (0.2) | 14 (0.2) | 12 (0.4) | | 24 (0.2) | 3 (0.3) | |
| | Harari | 26 (0.2) | 13 (0.2) | 13 (0.4) | | 25 (0.2) | 1 (0.1) | |
| | Addis Ababa | 244 (2.2) | 7 (0.1) | 237 (7.8) | | 189 (1.9) | 54 (5.7) | |
| | Dire Dwa | 47 (0.4) | 20 (0.3) | 27 (0.9) | | 42 (0.4) | 5 (0.5) | |
| Residence | Urban | 1216 (11.0) | 250 (3.1) | 966 (31.9) | 0.000 | 1003 (10.0) | 213 (22.6) | 0.000 |
| | Rural | 9807 (89.0) | 7747 (96.9) | 2060 (68.1) | | 9077 (90.0) | 730 (77.4) | |
| Family size | 1–3 | 1155 (10.5) | 607 (7.6) | 548 (18.1) | 0.000 | 1041 (10.3) | 115 (12.2) | 0.031 |
| | 4–6 | 5594 (50.7) | 3981 (49.8) | 1612 (53.3) | | 5097 (50.6) | 496 (52.6) | |
| | 7 and above | 4274 (38.8) | 3409(42.6) | 865 (28.6) | | 3942 (39.1) | 332 (35.2) | |

(*Continued*)

**Table 2.** (Continued)

| Birth order | First | 2058 (18.7) | 1036 (13.0) | 1022 (33.8) | 0.000 | 1835 (18.2) | 223 (23.7) | 0.000 |
|---|---|---|---|---|---|---|---|---|
| | 2–3 | 3359 (30.5) | 2334 (29.2) | 1025 (33.9) | | 3060 (30.4) | 299 (31.7) | |
| | 4–5 | 2604 (23.6) | 2102 (26.3) | 502 (16.6) | | 2403 (23.8) | 201 (21.3) | |
| | 6 and above | 3001 (27.2) | 2525 (31.6) | 476 (15.7) | | 2782 (27.6) | 219 (23.2) | |

| Variables | Categories | Weighted freque (%) | Place of Delivery | | | Postnatal checkup | | |
|---|---|---|---|---|---|---|---|---|
| | | | Home (%) | HF (%) | $X^2$-p value | No (%) | Yes (%) | $X^2$-p value |
| Birth interval | < = 24 | 2762 (25.1) | 2164 (27.1) | 598 (19.8) | 0.000 | 2567 (25.5) | 194 (20.6) | 0.000 |
| | 25–36 | 3510 (31.8) | 2679 (33.5) | 831 (27.5) | | 3208 (31.8) | 303 (32.1) | |
| | > = 37 | 4751 (43.1) | 3154 (39.4) | 1597 (52.8) | | 4305 (42.7) | 446 (47.3) | |
| Antenatal care | No | 7393 (67.1) | 6925 (68.7) | 468 (49.6) | 0.000 | 5935 (74.2) | 1440 (47.6) | 0.000 |
| | Yes | 3629 (32.9) | 3155 (31.3) | 475 (50.4) | | 2062 (25.8) | 1585 (52.4) | |
| Mass media | No | 7376 (66.9) | 5935 (74.2) | 1440 (47.6) | 0.000 | 6884 (68.3) | 492 (52.2) | 0.000 |
| | Yes | 3647 (33.1) | 2062 (25.8) | 1585 (52.4) | | 3196 (31.7) | 451 (47.8) | |
| Has mobile/ Telephone | No | 10851 (98.4) | 7916 (99.0) | 2935 (97.0) | 0.000 | 9923(98.5) | 928(98.4) | 0.92 |
| | Yes | 172(1.6) | 81 (1.0) | 91 (3.0) | | 156(1.5) | 15 (1.6) | |
| Wealth index | Poor | 5156 (46.8) | 4342(54.3) | 814 (26.9) | 0.000 | 4860 (48.2) | 296 (31.4) | 0.000 |
| | Middle | 2280 (20.7) | 1736 (21.7) | 544 (18.0) | | 2096 (20.8) | 184 (19.5) | |
| | Rich | 3587 (32.5) | 1919 (24.0) | 1668 (55.1) | | 3124 (31.0) | 463 (49.1) | |
| HF distance | Small problem | 6741 (61.2) | 4561 (57.0) | 2180 (72.1) | 0.000 | 6074 (60.3) | 666 (70.7) | 0.000 |
| | Big problem | 4281 (38.8) | 3436 (43.0) | 845 (27.9) | | 4005 (39.7) | 276 (29.3) | |
| Insurance | Not insured | 10633 (96.5) | 7794 (97.5) | 2839 (93.8) | 0.000 | 9751 (96.7) | 882 (93.5) | 0.000 |
| | Insured | 390 (3.5) | 203 (2.5) | 187 (6.2) | | 329 (3.3) | 61 (6.5) | |

Key: HF = health facility, SNNP = southern nation nationality people.

and not receive postnatal checkup were observed among newborns from poor household; 4342(54.3%) and 4860 (48.2%) respectively. In general, both DC and PNC utilization were more significantly associated with mother's education, mother's occupation, husband education, husband occupation, household head, religion, region, residence, family size, birth order, birth interval, antenatal care, mass media, wealth index, distance to health facility, and insurance; whereas only place of delivery was associated with mothers' age, marital status and has mobile/ telephone (Table 2).

## Bivariate analysis of socio-demographic and obstetric characteristics of women

The joint and marginal probabilities of DC and PNC together with the chi-square p-value was presented in (Table 3). Once the association between the two outcomes (DC and PNC) has been determined, the frequency distribution of each predictor for the different combinations of DC and PNC was done.

**Table 3. Joint and marginal probability of DC and PNC; EDHS 2016.**

| DC ($Y_1$) | PNC ($Y_2$) | | Marginal DC (%) | $X^2$ p-value |
|---|---|---|---|---|
| | No (%) | Yes (%) | | |
| Home (%) | 7498 (68.02) | 499 (4.53) | 7997 (72.55) | 0.000 |
| Health facility (%) | 2582 (22.42) | 444 (4.03) | 3026 (27.45) | |
| Marginal PNC (%) | 10080 (91.44) | 943 (8.56) | 11023 (100) | |

The frequency distributions of the combination of place of delivery (home and health center) and PNC (yes and no) with socio-demographic and obstetric variables is revealed in (Table 4). Women aged 25 to 29 years have a high proportion of PNC with a health center delivery (approximately 31.8%), whereas the majority of women across all age groups gave birth at home and had no PNC. Women who did not attend formal education and had primary education accounted for the majority of home delivery with no PNC utilization; approximately 5677 (75.7%) and 1702(22.7%) respectively. Regarding mothers' occupation, more than half of working women deliver at health center followed by postnatal checkup; on the other side about 4341(57.9) of women who have no job deliver at their home and did not attend PNC. Delivery at health center with PNC were well practiced at Amhara region 118(26.5%) followed by SNNP 86(19.3%), Oromia 83(18.7%), Tigray 81(18.2%), and Addis Ababa 53(11.9%), in contrast with this, approximately half of women who had deliver at home without PNC were also recorded at Oromia region. Delivery care followed by postnatal care utilization was more practiced among women who had a higher wealth index compared to those with medium and lower wealth index. Moreover, lower wealth indexed women had deliver at home and had not attended postnatal checkup (Table 4).

## Regional prevalence of home delivery and no PNC

Fig 1 displays the prevalence of home delivery and not receive postnatal checkup within two months after birth across regions of Ethiopia. Even though the proportion of both home delivery and no PNC were high in each region, the later one was more serious problem. The highest prevalence of home delivery and no PNC were observed in Afar (85.1%) and Harari (96.2%) region respectively. Whereas the smallest proportion of both home delivery and no PNC were recorded in Addis Ababa (2.9% and 77.8% respectively) (Fig 1).

## Spatial analysis

**Spatial autocorrelation of DC and PNC.** The estimated Global Moran's I statistic for home delivery and no PNC were 0.378 and 0.177 respectively, with a p-value of <0.001, indicates that the spatial distribution of DC and PNC was significantly clustered across EAs. Hence, geographically close EAs are more related than distant areas (Table 5).

**Spatial distribution.** The proportion of home delivery and no postnatal checkup of women-child pair in each enumeration areas was represented by different colors; points with red color indicates enumeration areas with high proportion of home delivery (left) and high no PNC (right), and points with green color shows areas that had a high proportion of health center delivery (left) and high proportion of postnatal checkup (right), (Fig 2).

**Hot spot analysis.** A point with red color indicates significant hot spot areas of home delivery (left) and no PNC (right); more specifically high home delivery was observed around Gondar, North Wollo, Afar-Zone1, 3 and 4, south wollo, Argoba special wereda, North Shewa, Metekel, Awi Agew, east Gojjam, Nuer, Jimma, Gurage, Silte, Alaba, Yem, Wolayita, Gamo Gofa, Sidama, Gedio, Fafan, Jarar, Korahe and Doolo, and no PNC was observed around south Gondar, east Gojjam, Afar-zone1, Dire Dawa, east Harargie, Fafan, Jarar, Korahe, Shebelle, Nuer, Jimma, Sheka, and Keffa (Fig 3).

**Spatial interpolation.** The spatial kriging interpolation analysis was used to predict home delivery and baby not receive postnatal checkup within two months after birth at non-sampled areas of the country. The high and low predicted areas of home delivery and baby not receive postnatal checkup within two months after birth was indicated by red and green colors respectively in Fig 4. The Liben, Borena, Guji, Bale, Dolo and Zone 2 were predicted to have relatively

**Table 4. Frequency distribution of socio-demographic and obstetric characteristics for the different combinations of DC and PNC; EDHS 2016.**

| Variables | Categories | Home and No PNC (%) | Home and PNC (%) | HF and No PNC (%) | HF and PNC (%) |
|---|---|---|---|---|---|
| Age | 15–19 | 214 (2.9) | 9 (1.8) | 132 (5.1) | 23 (5.2) |
| | 20–24 | 1271 (17.0) | 68 (13.6) | 621 (24.1) | 107 (24.2) |
| | 25–29 | 2239 (29.9) | 169 (33.9) | 804 (31.1) | 141 (31.8) |
| | 30–34 | 1771 (23.6) | 118 (23.6) | 517 (20.0) | 84 (19.0) |
| | 35–39 | 1263 (16.8) | 68 (13.6) | 381 (14.8) | 60 (13.5) |
| | 40–44 | 547 (7.3) | 52 (10.4) | 101 (3.9) | 22 (5.0) |
| | 45–49 | 193 (2.6) | 15 (3.0) | 26 (1.0) | 6 (1.4) |
| Mother's education | No education | 5677 (75.7) | 365 (73.3) | 1084 (42.0) | 158 (35.7) |
| | Primary | 1702 (22.7) | 117 (23.5) | 970 (37.6) | 161 (36.3) |
| | Secondary | 97 (1.3) | 15 (3.0) | 326 (12.6) | 76 (17.2) |
| | Higher | 22 (0.3) | 1(0.2) | 203 (7.9) | 48 (10.8) |
| Mother's Occupation | Not working | 4341 (57.9) | 265 (53.1) | 1321 (51.2) | 199 (44.9) |
| | Working | 3157 (42.1) | 234 (46.9) | 1260 (48.8) | 244 (55.1) |
| Husband education | No education | 4108 (54.8) | 250 (50.0) | 873 (33.8) | 126 (28.4) |
| | Primary | 2971 (39.6) | 210 (42.0) | 954 (37.0) | 169 (38.1) |
| | Secondary | 317 (4.2) | 28 (5.6) | 431 (16.7) | 76 (17.2) |
| | Higher | 102 (1.4) | 12 (2.4) | 323 (12.5) | 72 (16.3) |
| Husband occupation | Not working | 708 (9.4) | 39 (7.8) | 186 (7.2) | 27 (6.1) |
| | Working | 6790 (90.6) | 460 (92.2) | 2396 (92.8) | 417 (93.9) |
| Household head | Male | 6553 (87.4) | 433 (86.8) | 2154 (83.4) | 353 (79.7) |
| | Female | 945 (12.6) | 66 (13.2) | 428 (16.6) | 90 (20.3) |
| Marital status | Unmarried | 347 (4.6) | 21 (4.2) | 160 (6.2) | 32 (7.2) |
| | Married | 7151 (95.4) | 478 (95.8) | 2422 (93.8) | 412 (92.8) |
| Religion | Orthodox | 2177 (29.0) | 162 (32.4) | 1167 (45.2) | 266 (59.9) |
| | Catholic | 76 (1.0) | 7 (1.4) | 17 (0.7) | 4 (0.9) |
| | Protestant | 1612 (21.5) | 121 (24.2) | 520 (20.1) | 77 (17.3) |
| | Muslim | 3400 (45.3) | 205 (41.0) | 860 (33.3) | 97 (21.8) |
| | Others | 233 (3.1) | 5 (1.0) | 19 (0.7) | 0 (0.0) |
| Region | Tigray | 256 (3.4) | 38 (7.6) | 341 (13.2) | 81 (18.2) |
| | Afar | 93 (1.2) | 4 (0.8) | 14 (0.5) | 3 (0.7) |
| | Amhara | 1376 (18.4) | 102 (20.4) | 476 (18.4) | 118 (26.5) |
| | Oromo | 3700 (49.4) | 203 (40.7) | 865 (33.5) | 83 (18.7) |
| | Somalia | 393 (5.2) | 23 (4.6) | 85 (3.3) | 7 (1.6) |
| | Benishangul | 81 (1.1) | 9 (1.8) | 25 (1.0) | 8 (1.8) |
| | SNNP | 1549 (20.7) | 115 (23.0) | 546 (21.1) | 86 (19.3) |
| | Gambela | 13 (0.2) | 1 (0.2) | 11 (0.4) | 2 (0.4) |
| | Harari | 12 (0.2) | 1 (0.2) | 12 (0.5) | 1 (0.2) |
| | Addis Ababa | 6 (0.1) | 1 (0.2) | 184 (7.1) | 53 (11.9) |
| | Dire Dwa | 18 (0.2) | 2 (0.4) | 24 (0.9) | 3 (0.7) |
| Residence | Urban | 214 (2.9) | 36 (7.2) | 789 (30.6) | 177 (39.9) |
| | Rural | 7284 (97.1) | 463 (92.8) | 1793 (69.4) | 267 (60.1) |
| Family size | 1–3 | 584 (7.8) | 24 (4.8) | 457 (17.7) | 92 (20.7) |
| | 4–6 | 3743 (49.9) | 238 (47.6) | 1354 (52.4) | 258 (58.1) |
| | 7 and above | 3171 (42.3) | 238 (47.6) | 771 (29.9) | 94 (21.2) |
| Birth order | First | 986 (13.2) | 49 (9.8) | 849 (32.9) | 174 (39.2) |
| | 2–3 | 2179 (29.1) | 155 (31.0) | 881 (34.1) | 144 (32.4) |
| | 4–5 | 1958 (26.1) | 145 (29.0) | 445 (17.2) | 57 (12.8) |
| | 6 and above | 2374 (31.7) | 151 (30.2) | 407 (15.8) | 69 (15.5) |

(*Continued*)

**Table 4.** (Continued)

| Variables | Categories | Home and No PNC (%) | Home and PNC (%) | HF and No PNC (%) | HF and PNC (%) |
|---|---|---|---|---|---|
| Birth interval | < = 24 | 2050 (27.3) | 114 (22.8) | 517 (20.0) | 81 (18.2) |
| | 25–36 | 2497 (33.3) | 183 (36.6) | 711 (27.5) | 120 (27.0) |
| | > = 37 | 2951 (39.4) | 203 (40.6) | 1354 (52.4) | 243 (54.7) |
| Antenatal care | No | 5660 (75.5) | 297 (59.5) | 1266 (49.0) | 170 (38.4) |
| | Yes | 1838 (24.5) | 202 (40.5) | 1316 (51.0) | 273 (61.6) |
| Mass media | No | 5613 (74.9) | 322 (64.4) | 1270 (49.2) | 170 (38.3) |
| | Yes | 1885 (25.1) | 178 (35.6) | 1311 (50.8) | 274 (61.7) |
| Has mobile/ Telephone | No | 7419 (98.9) | 497 (99.6) | 2505 (97.0) | 430 (97.1) |
| | Yes | 79 (1.1) | 2 (0.4) | 77 (3.0) | 13 (2.9) |
| Wealth index | Poor | 4123 (55.0) | 219 (43.8) | 737 (28.5) | 77 (17.4) |
| | Middle | 1628 (21.7) | 108 (21.6) | 467 (18.1) | 76 (17.2) |
| | Rich | 1746 (23.3) | 173 (34.6) | 1378 (53.4) | 290 (65.5) |
| HF distance | Small problem | 4254 (56.7) | 306 (61.3) | 1820 (70.5) | 360 (81.1) |
| | Big problem | 3244 (43.3) | 193 (38.7) | 762 (29.5) | 84 (18.9) |
| Insurance | Not insured | 7316 (97.6) | 478 (95.6) | 2435 (94.3) | 404 (91.2) |
| | Insured | 182 (2.4) | 22 (4.4) | 147 (5.7) | 39 (8.8) |

Key: HF = health facility, SNNP = southern nation nationality people.

high home delivery. Similarly part of Afder, Shabelle, Korahe, Dolo and Zone 2 had high no postnatal checkup within 2 months after birth (Fig 4).

## Model fitting and parameters estimation

**Bivariate binary with spatial effect model for DC and PNC.** Table 6 depicts simultaneous effect of socio-demographic and obstetrics covariates on DC and PNC. The odds ratio of 0.029 confirms the dependency between DC and PNC, having this result a spatial bivariate binary logistic regression model was applied to analysis the effect of each predictor on DC and PNC.

Holding all other predictors in the model, the odds of home delivery was 1.68, 1.92, 1.75, 1.68 and 1.79 times higher among women aged 20–24, 25–29, 30–34, 35–39, and 40–44 years respectively than women aged 15–19 years. Likewise, Compared to women without any formal

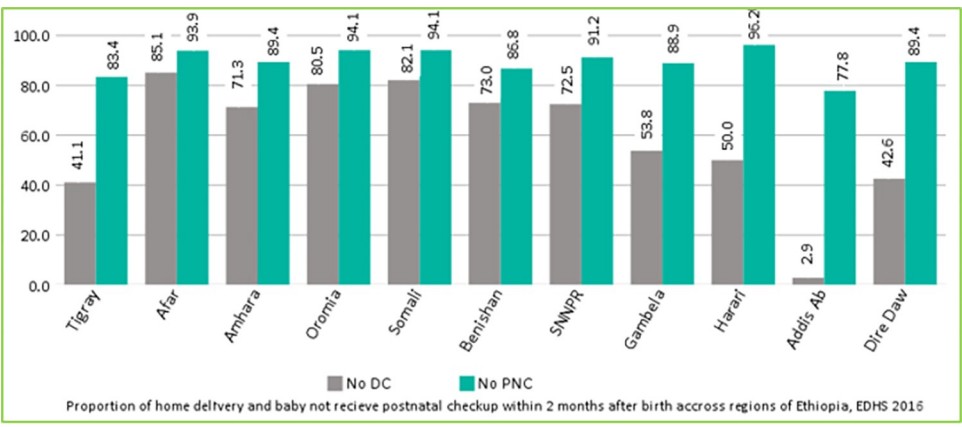

**Fig 1. Prevalence of home delivery and no PNC across regions of Ethiopia; EDHS 2016.**

**Table 5. Indicator of spatial autocorrelation.**

| Dependent variable | Moran's I statistic | Z-score | P-value |
|---|---|---|---|
| No DC | 0.378 | 49.96 | < 0.001 |
| **No PNC** | **0.177** | **23.62** | **< 0.001** |

Key: No DC = No delivery care; No PNC = No postnatal checkup.

education, women with primary, secondary, and higher education were 39%, 62%, and 70% less likely to deliver at home, respectively.

Adjusting all other variables, in comparison to woman with a non-educated husband, a woman who had a secondary and higher educated partner was 48% and 35% less likely to deliver at home respectively. Regarding to women occupation, the odds that a baby born from a working mother not attended a PNC was lower by 22% compared to babies born from a non-working mother did not received a PNC.

The findings of this study also reviled that, compared to Tigray region, the odds that mothers from Afar, Oromia, Somali, Benishangul-Gumu, SNNP, Gambela, Harari and Addis Ababa regions deliver at home were 10.59, 8.069, 6.05, 3.13, 2.51, 1.97, 2.34 and 1.96 times higher respectively. Similarly; the odds that a baby born from a mother reside in Afar, Oromia, Somali, SNNP, Gambela, Harari, and Dire Dawa not receive postnatal checkup were 2.16, 2.10, 3.60, 1.12, 1.63, 1.49, 5.64 and 2.25 times the odds that a baby born from a mother reside in Tigray not receive postnatal checkup respectively.

Also, keeping other predictors; women who had attended the recommended antenatal care visit during pregnancy were 52% and 66% less likely to deliver at home and not follow up postnatal checkup respectively compared to women not attended antenatal care. Concerning

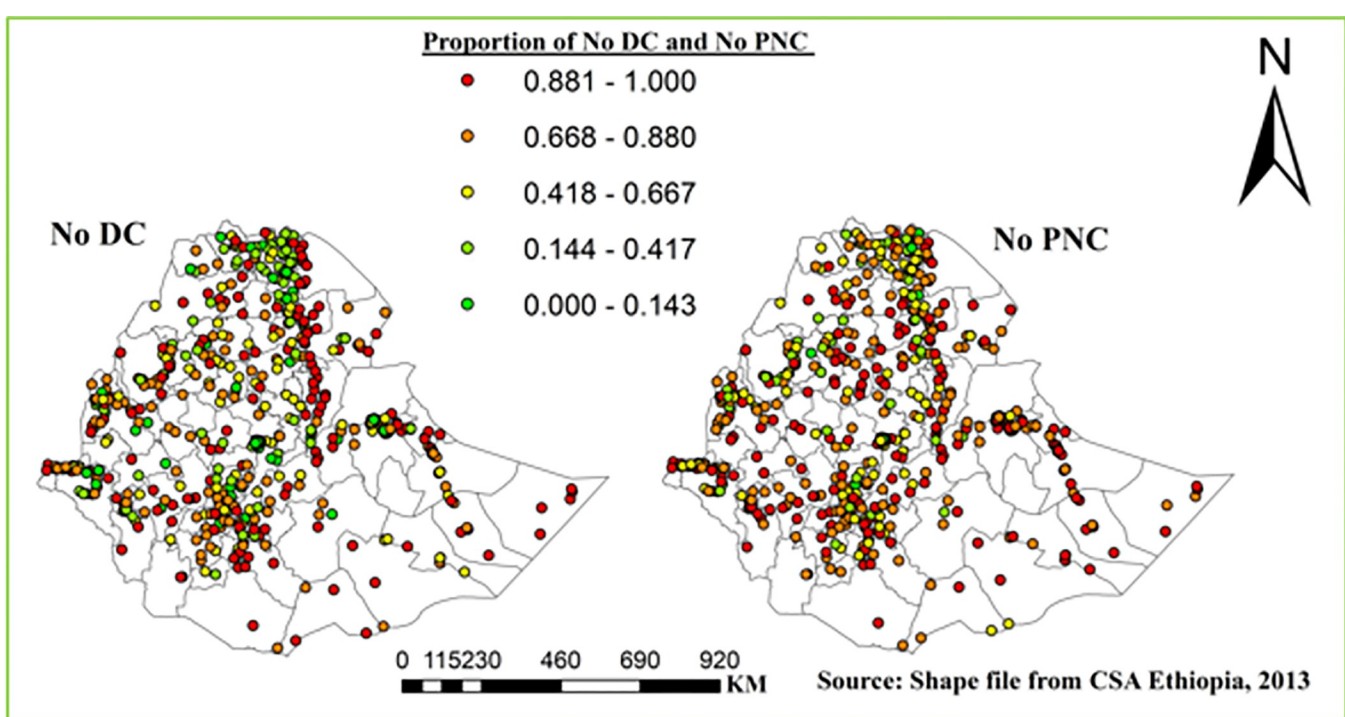

**Fig 2. Spatial distribution of DC and PNC in Ethiopia, EDHS 2016.**

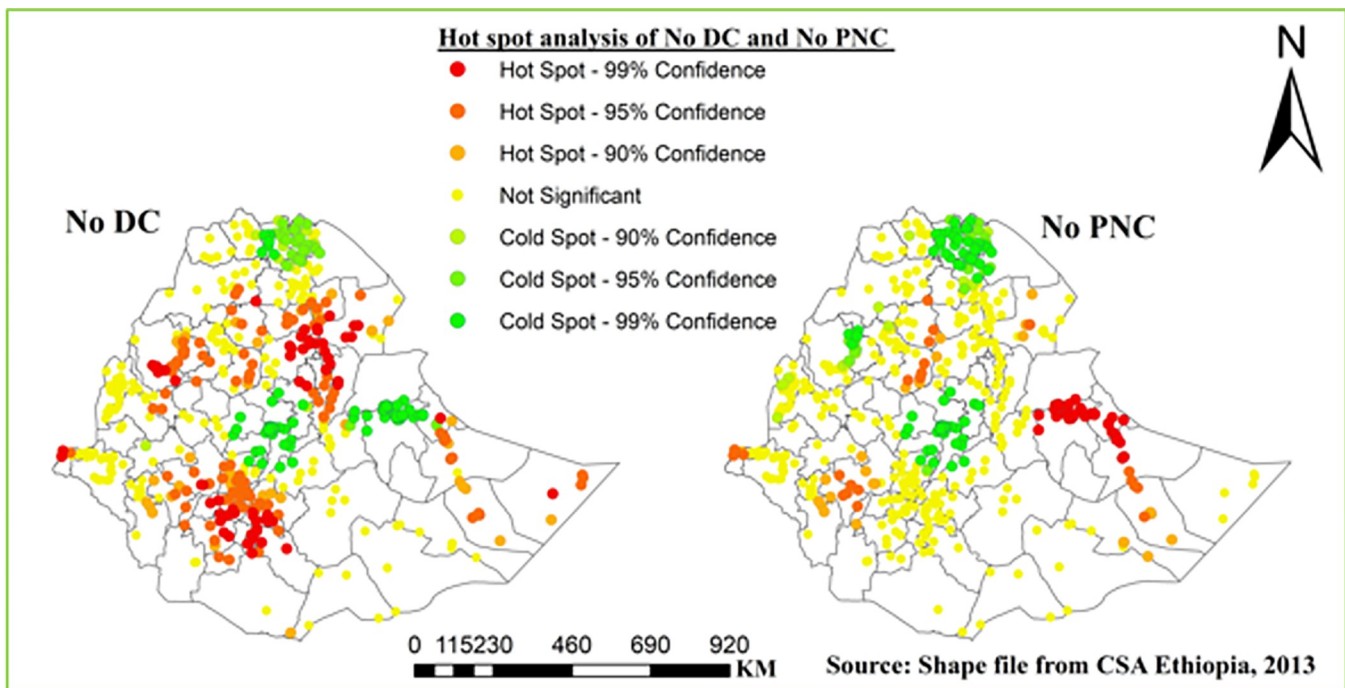

**Fig 3. Hotspot analysis of DC and PNC in Ethiopia, EDHS 2016.**

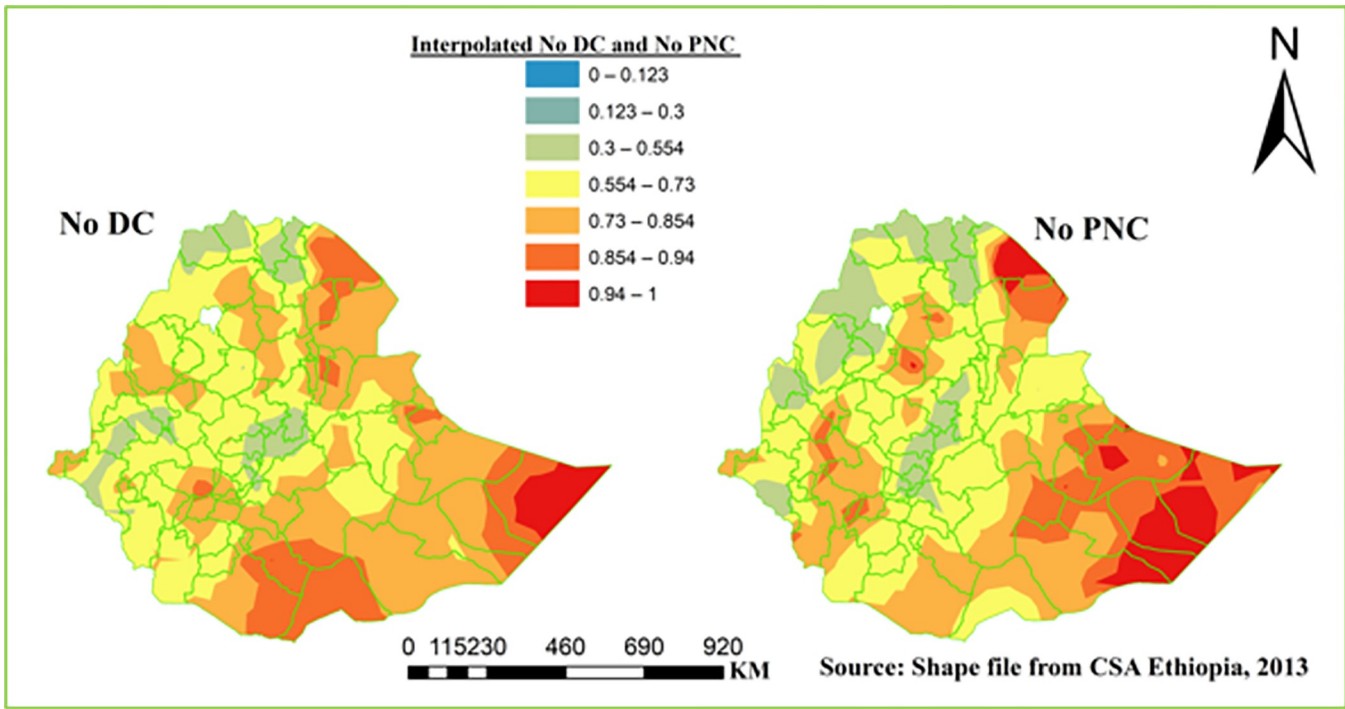

**Fig 4. Spatial interpolation of DC and PNC in Ethiopia, EDHS 2016.**

**Table 6. Parameter estimates of the spatial bivariate binary logistic regression modeling of DC and PNC; EDHS 2016.**

| Variables | Place of Delivery (event = Home) | | PNC (event = No) | |
|---|---|---|---|---|
| | Estimate (se) | AOR (95% CI) | Estimate (se) | AOR (95% CI) |
| **Intercept** | -1.22 (0.55) | – | 5.95 (2.035) | – |
| **Age** (ref = 15–19) | | | | |
| 20–24 | 0.52 (0.17) | 1.68 (1.21, 2.35) | -0.12 (0.249) | 0.89 (0.54, 1.44) |
| 25–29 | 0.65 (0.181) | 1.92 (1.34, 2.73) | -0.35 (0.260) | 0.70 (0.42, 1.17) |
| 30–34 | 0.56 (0.198) | 1.75 (1.19, 2.58) | -0.40 (0.281) | 0.67 (0.39, 1.16) |
| 35–39 | 0.52 (0.212) | 1.68 (1.11, 2.56) | -0.42 (0.301) | 0.66 (0.36, 1.18) |
| 40–44 | 0.58 (0.252) | 1.79 (1.091, 2.93) | -0.16 (0.368) | 0.85 (0.42, 1.79) |
| 45–49 | 0.29 (0.335) | 1.34 (0.69, 2.58) | -0.71 (0.441) | 0.49 (0.21, 1.17) |
| **Mother's education**(ref = No edu) | | | | |
| Primary | -0.50 (0.085) | 0.61 (0.51, 0.72) | -0.15 (0.119) | 0.86 (0.68, 1.090) |
| Secondary | -0.98 (0.152) | 0.38 (0.28, 0.51) | -0.068 (0.203) | 0.93 (0.63, 1.39) |
| Higher | -1.20 (0.252) | 0.30 (0.18, 0.50) | -0.12 (0.276) | 0.89 (0.52, 1.52) |
| **Mother Occup** (ref = Not working) | | | | |
| Working | -0.032 (0.0747) | 0.97 (0.84, 1.12) | -0.25 (0.0993) | 0.78 (0.64, 0.94) |
| **Husband education** (ref = No edu) | | | | |
| Primary | -0.046 (0.0839) | 0.96 (0.81, 1.13) | -0.066 (0.113) | 0.94 (0.75, 1.17) |
| Secondary | -0.65 (0.120) | 0.52 (0.41, 0.66) | -0.17 (0.164) | 0.84 (0.61, 1.17) |
| Higher | -0.43 (0.145) | 0.65 (0.49, 0.86) | 0.0067 (0.200) | 1.00 (0.68, 1.49) |
| **Husband occup** (ref = Not working) | | | | |
| Working | -0.032 (0.108) | 0.97 (0.78, 1.20) | -0.269 (0.158) | 0.76 (0.56, 1.041) |
| **Religion** (ref = Orthodox) | | | | |
| Catholic | 0.45 (0.373) | 1.57 (0.75, 3.25) | 0.66 (0.617) | 1.93 (0.58, 6.46) |
| Protestant | 0.63 (0.142) | 1.88 (1.42, 2.47) | 0.058 (0.182) | 1.0 (0.74, 1.51) |
| Muslim | -0.21 (0.125) | 0.81 (0.63, 1.031) | 0.24 (0.168) | 1.27 (0.56, 1.090) |
| Others | 1.022 (0.355) | 2.78 (1.38, 5.57) | -0.18 (0.352) | 0.84 (0.42, 1.67) |
| **Region** (ref = Tigray) | | | | |
| Afar | 2.36 (0.193) | 10.59 (7.29, 15.53) | 0.77 (0.235) | 2.16 (1.36, 3.41) |
| Amhara | 0.61 (0.314) | 1.84 (0.99, 3.39) | -0.47 (0.350) | 0.63 (0.31, 1.23) |
| Oromo | 2.088 (0.194) | 8.069 (5.52, 11.90) | 0.74 (0.232) | 2.10 (1.32, 3.29) |
| Somalia | 1.80 (0.192) | 6.050 (4.16, 8.84) | 1.28 (0.248) | 3.60 (2.20, 5.82) |
| Benishangul | 1.14 (0.156) | 3.13 (2.30, 4.24) | 0.11 (0.192) | 1.12 (0.77, 1.63) |
| SNNP | 0.92 (0.164) | 2.51 (1.82, 3.45) | 0.49 (0.209 | 1.63 (1.08, 2.44) |
| Gambela | 0.68 (0.190) | 1.97 (1.37, 2.87) | 0.40 (0.234) | 1.49 (1.06, 2.35) |
| Harari | 0.85 (0.188) | 2.34 (1.62, 3.38) | 1.73 (0.278) | 5.64 (3.28, 9.78) |
| Addis Ababa | 0.80 (0.153) | 1.69 (1.28, 2.22) | 0.71 (0.137) | 2.25 (1.75, 2.88) |
| Dire Dawa | 0.05(0.130) | 1.05 (0.82, 1.36) | -0.12 (0.110) | 0.89 (0.72, 1.10) |
| **Household head** (ref = male) | | | | |
| Female | -0.01 (0.091) | 0.99 (0.83, 1.19) | -0.095 (0.118) | 0.91 (0.72, 1.15) |
| **Marital status** (ref = Unmarried) | | | | |
| Married | 0.28 (0.149) | 1.32 (0.99, 1.78) | 0.27 (0.189) | 1.31 (0.90, 1.89) |
| **Residence** (ref = Urban) | | | | |
| Rural | 1.30 (0.108) | 3.67 (2.99, 4.57) | 0.03 (0.154) | 1.03 (0.76, 1.39) |
| **Family size** (ref = 1–3) | | | | |
| 4–6 | 0.29 (0.17) | 1.34 (1.07, 1.69) | -0.47 (0.16) | 0.63 (0.46, 0.86) |
| 7 and above | 0.33 (0.13) | 1.39 (1.07, 1.79) | -0.53 (0.19) | 0.59 (0.41, 0.84) |

*(Continued)*

**Table 6.** (Continued)

| Variables | Place of Delivery (event = Home) | | PNC (event = No) | |
| --- | --- | --- | --- | --- |
| | Estimate (se) | AOR (95% CI) | Estimate (se) | AOR (95% CI) |
| Variables | Place of Delivery (event = Home) | | PNC (event = No) | |
| | Estimate (se) | AOR (95% CI) | Estimate (se) | AOR (95% CI) |
| **Birth order (ref** = First) | | | | |
| 2–3 | 0.25 (0.11) | 1.28 (1.03, 1.60) | 0.37 (0.14) | 1.45 (1.09, 1.90) |
| 4–5 | 0.47 (0.14) | 1.60 (1.22, 2.10) | 0.68 (0.18) | 1.98 (1.39, 2.81) |
| 6 and above | 0.19 (0.17) | 1.21 (0.87, 1.67) | 0.68 (0.22) | 1.97 (1.29, 3.02) |
| **Birth interval** (ref = < = 24) | | | | |
| 25–36 | -0.004 (0.09) | 0.99 (0.84, 1.19) | 0.15 (0.12) | 1.16 (0.93, 1.47) |
| > = 37 | -0.20 (0.09) | 0.82 (0.69, 0.97) | (0.22 (0.12) | 1.25 (0.99, 1.55) |
| **Antenatal care** (ref = No) | | | | |
| Yes | -0.73 (0.07) | 0.48 (0.42, 0.55) | -0.44 (0.09) | 0.64 (0.53, 0.77) |
| **Access to mass media** (ref = No) | | | | |
| Yes | -0.37 (0.08) | 0.69 (0.59, 0.81) | -0.43 (0.12) | 0.65 (0.53, 0.80) |
| **Has mobile/ Telephone** (ref = No) | | | | |
| Yes | -0.64 (0.33) | 0.53 (0.28, 1.001) | 0.41 (0.40) | 1.51 (0.70, 3.29) |
| **Wealth index** (ref = Poor) | | | | |
| Middle | -0.37 (0.10) | 0.69 (0.57, 0.84) | -0.24 (0.14) | 0.79 (0.60, 1.03) |
| Rich | -0.69 (0.10) | 0.50 (0.42, 0.60) | -0.12 (0.14) | 0.89 (0.68, 1.15) |
| **HF distance** (ref = Not big problem) | | | | |
| Big problem | 0.30 (0.08) | 1.35 (1.16, 1.59) | 0.24 (0.110) | 1.27 (1.03, 1.58) |
| **Insurance** (ref = Not insured) | | | | |
| Insured | -0.33 (0.26) | 0.72 (0.43, 1.20) | -0.19 (0.29) | 0.83 (0.47, 1.47) |
| Auto covariance (Si) | 0.48 (0.21) | 1.62 (1.07, 2.44) | 0.81(0.27) | 2.25 (1.32, 3.82) |
| **Measure of Dependency:** Odds Ratio (OR) 0.029 | | | | |

Key: ref = reference, No edu = no education, SNNP = southern nation nationality people, HF distance = distance to health facility.

wealth status, the odds that a mother from medium and rich household delivered at home were lower by 31% and 50% respectively as compared to the odds that a mother reside in a poor household deliver at home. Moreover, the odds of not utilizing delivery care and postnatal checkup among women who had faced a big problem of health facility distance were 35% and 27% greater than the odds that a women who had not big problem of health facility distance deliver at home and not follow up postnatal checkup receptively. The spatial variable (auto covariance) with a positive coefficient (0.48 and 0.81 for DC and PNC respectively) incorporates that zones with high prevalence of DC and/or PNC were surrounded by zones with high prevalence of DC and/or PNC and vice versa (Table 6).

## Discussion

The aim of this study was to investigate geographic variations and determinants of low utilization of delivery care services and postnatal check-ups within 2 months following live births in Ethiopia. The study found that 72.55% of women give birth at home, which is consistent with previous studies from Southeast Ethiopia (73.6%) [31], EDHS 2016 report (73.3%) [32], and Gozamin District in Amhara region (75.3%) [33], but higher than studies from Afar, Ethiopia (65%), South Ethiopia (62.2%), and Arba Minch town Ethiopia (33.2%) [34–36], Zala woreda,

Southern Ethiopia (67.6%) [13], Anlemo District, Hadiya Zone (49.3%) [1], and Wolaita and Dawro Zone (62%) [1,14,37,38]; in contrast it is lower than a study conducted in Arbaminch Zuria District (79.4%) [39], zone 3 of Afar region (83.3%) [40]. In terms of PNC following deliveries, around 91.45% of women did not have a PNC after giving birth; this finding is consistent with studies conducted in Ethiopia [41], but slightly higher and lower than studies conducted in the same country [42,43] respectively. This variation could be attributable to the study area, setting difference, cultural attitude towards health facility delivery, infrastructure difference (access to the health facility, roads, and distance to the health facility).

According to the study's findings, ANC visits were substantially associated with place of delivery and PNC. Mothers who attended an ANC visit were less likely to give birth at home and not utilize PNC than mothers who did not attend an ANC visit. Similar evidences of greater magnitude have been reported from researches conducted in different parts of Ethiopia [21,41,44–47], Akordet town, Eritrea [48], Uganda [49], and Tanzania [50]. This could be related to women's awareness/knowledge about DC and PNC during an ANC visit [51].

In pursuance of mother's age, older women were more likely to deliver at home compared to younger. This result is in agreement with studies conducted in Ethiopia, Zambia, Tanzania and Nepal [52–55]. The possible justification may be due to the fact that older women may consider themselves as experienced (may have more than one birth earlier) and no need to have assistance from skilled health professionals.

Mother's place of delivery was also significantly influenced by her level of education. Women with a primary, secondary, or higher education were less likely to give birth at home than those with no education, which was consistent with the findings of other studies conducted in different regions of Ethiopia [39,56,57]. Because education increases women's comprehension and awareness of the benefits of health care utilization and difficulties during pregnancy and childbirth, it strengthens women's habits of deciding where and how to access the better health care services.

Women whose husbands had secondary or higher education were more likely to give birth at a health facility than women whose husbands had no education. The results were in line with a prior investigation carried out in the Oromiya region [57]. This may be due to fact that educated spouses might be more accepting of contemporary medicine, aware of benefits of health care utilization and the advantages of giving birth at health facility.

Similar to other studies in Ethiopia, Ghana and Nigeria [21,51,58–61] regarding delivery care and in Ethiopia [19] regarding PNC, the finding of our study found that distance to health facility was one of the influencing factors that hinder women from accessing health facility delivery care and PNC. As a result, rural women were more likely than urban women to give birth at home. Ethiopian studies at the national and regional levels had consistent results with our findings [21,52,62]. This could be because, as compared to urban women, rural women are less likely to be aware of the benefits of using health care and giving birth in a health facility, and have no or limited access to maternal health care (due to infrastructure, distance from health facility, spouse knowledge/attitude, and cultural behaviors towards health facility delivery).

According to this study, having a low wealth status increases the likelihood of delivering at home compared to having a medium or higher economic position. This outcome was consistent with studies undertaken in Ethiopia and Bangladesh [21,52,63–65]. This could be because women assume that they cannot cover the cost of services; yet, exempted delivery services and transportation are available. This demonstrates that the community is unaware of the provision; as a result, the government must seek to raise awareness and motivate the community to use the free delivery services and transportation provided.

There is a significant association between birth order and place of delivery; women with high birth order (2 or higher) were more likely to deliver at home than women with first birth, which is consistent with other studies [58,64,65]. This could be because mothers with high birth orders have experience of managing home deliveries with or without the assistance of community midwives.

## Strength and limitation

The big dataset from the EDHS survey and the study's national representativeness are its strong points, and these factors contribute to its adequate statistical power. Rigorous statistical analyses, including spatial analysis, were conducted to identify hotspot and cold spot areas, adding further depth to the findings. Despite the advantages listed above, the study has the following drawback. Given that the events occurred five years prior to the survey and the majority of the data was dependent on the mother's response, there may be recall bias, which could result from an inability to recall some of the traits that were examined. We used the EDHS 2016, which was collected seven years before to the study's execution, because the most recent survey data (Mini EDHS 2019) does not fully describe the issues of study (delivery care and postnatal check-up). As a result, the findings may not accurately reflect Ethiopia's current postnatal checkup and delivery care service utilization.

## Conclusion

This study found that more than 72% and 91% of women delivered at home and do not attend postnatal check-up within two months following birth respectively. The spatial distribution of DC and PNC were significantly clustered across EAs, implying that space had an effect on both DC and PNC. Low utilization of both DC and PNC were highly observed around Liben, Borena, Guji, Bale, Dolo, Zone 2, Afder, Shabelle, and Korahe.

Lack of occupation, region, large family size, higher birth order, low utilization of antenatal care visit, unable to access mass media, big problem of health facility distance and the spatial variable were found to be jointly significant predictors of low utilization of DC and PNC in Ethiopia. Whereas older age, being reside in rural area and low wealth status affects delivery care service utilization.

Low utilizations of delivery care service and postnatal check-up are major cause of maternal and infant death; hence, we suggest the government, stakeholders, health providers and policymakers give emphasis to maternal health services (such as delivery and post-delivery cares) by considering those variables. We also recommend researchers to conduct further studies using the latest survey data set to investigate the current utilization of DC and PNC.

## Acknowledgments

We are thank full to measure DHS program for data availability.

## Author Contributions

**Conceptualization:** Frezer Tilahun Getaneh, Muluwerk Ayele Derebe.

**Data curation:** Shegaw Mamaru Awoke.

**Formal analysis:** Shegaw Mamaru Awoke.

**Funding acquisition:** Shegaw Mamaru Awoke.

**Investigation:** Shegaw Mamaru Awoke.

**Methodology:** Shegaw Mamaru Awoke.

**Project administration:** Frezer Tilahun Getaneh.

**Resources:** Frezer Tilahun Getaneh.

**Software:** Shegaw Mamaru Awoke.

**Supervision:** Muluwerk Ayele Derebe.

**Validation:** Shegaw Mamaru Awoke, Frezer Tilahun Getaneh, Muluwerk Ayele Derebe.

**Visualization:** Shegaw Mamaru Awoke, Frezer Tilahun Getaneh, Muluwerk Ayele Derebe.

**Writing – original draft:** Shegaw Mamaru Awoke, Frezer Tilahun Getaneh.

**Writing – review & editing:** Shegaw Mamaru Awoke, Frezer Tilahun Getaneh, Muluwerk Ayele Derebe.

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
