## [Decision Letter · Decision Letter 0]

20 Sep 2023

PONE-D-23-23873Spatial Distribution of Home Delivery and Baby not Receive Postnatal Check-up within 2 Months after Birth in Ethiopia: Bivariate AnalysisPLOS ONE

Dear Dr. Awoke,

Thank you for submitting your manuscript to PLOS ONE. After careful consideration, we feel that it has merit but does not fully meet PLOS ONE’s publication criteria as it currently stands. Therefore, we invite you to submit a revised version of the manuscript that addresses the points raised during the review process.

We look forward to receiving your revised manuscript.

Kind regards,

Gulzhanat Aimagambetova

Academic Editor

PLOS ONE

Journal Requirements:

4. We note that Figures 2, 3 and 4 in your submission contain map images which may be copyrighted. All PLOS content is published under the Creative Commons Attribution License (CC BY 4.0), which means that the manuscript, images, and Supporting Information files will be freely available online, and any third party is permitted to access, download, copy, distribute, and use these materials in any way, even commercially, with proper attribution. For these reasons, we cannot publish previously copyrighted maps or satellite images created using proprietary data, such as Google software (Google Maps, Street View, and Earth). For more information, see our copyright guidelines: http://journals.plos.org/plosone/s/licenses-and-copyright.

We require you to either present written permission from the copyright holder to publish these figures specifically under the CC BY 4.0 license, or (2) remove the figures from your submission:

a. You may seek permission from the original copyright holder of Figures 2, 3 and 4 to publish the content specifically under the CC BY 4.0 license.  

Reviewers' comments:

Reviewer's Responses to Questions

**Comments to the Author**

1. Is the manuscript technically sound, and do the data support the conclusions?

Reviewer #1: Yes

Reviewer #2: Yes

2. Has the statistical analysis been performed appropriately and rigorously? 

Reviewer #1: Yes

Reviewer #2: I Don't Know

3. Have the authors made all data underlying the findings in their manuscript fully available?

Reviewer #1: Yes

Reviewer #2: Yes

4. Is the manuscript presented in an intelligible fashion and written in standard English?

Reviewer #1: Yes

Reviewer #2: No

5. Review Comments to the Author

Reviewer #1: A good effort's to highlight this topic by providing with detailed statistical analysis,but there are many already provided literatures which focuses on postnatal checkup and delivery care,. This study cannot add much new information to the existing literature.

Reviewer #2: � Comments to Authors:

Title:

• The title of the study is relatively good. However, the authors should revise the title to be more inclusive and precise to the study’s objectives and aims (such as the determinants of PNC utilization).

Introduction:

• It is clear, concise, and straightforward, and the rationale and aim of the study are well-stated. In addition, important definitions were illustrated clearly.

Methods:

• The authors conducted a database study and obtained ethical approval as requested.

• Sampling method was illustrated properly.

Results:

• The authors conducted a thorough analysis of the results, supported by sufficient amount of data justifying the conclusion. Nevertheless, the analysis assumed a linear relationship between dependent and independent variables, overlooking potential confounding factors that could impact the accuracy of correlations. Therefore, further research may be warranted to validate these hypotheses.

Discussion:

• In the discussion section, the authors have effectively compared their findings with those of similar studies and offered reasonable explanations for both the similarities and differences. It is crucial, however, to avoid over interpreting the data.

• While the authors did highlight the clinical implications of their study's findings, it is worth noting that they missed addressing the study's limitations. It is advisable to have a transparent discussion of these limitations to provide a more comprehensive understanding of the research's scope and potential constraint.

• Further clinical application should be mentioned for policy makers, together with future direction and recommendation.

Conclusion:

• Correctly answered the research question.

• Authors should consider conciseness of conclusion to represent the whole findings.

Minor:

• English editing and proofreading by a native English speaker needed. The authors should correct some typos and language mistakes.

6. PLOS authors have the option to publish the peer review history of their article (what does this mean?). If published, this will include your full peer review and any attached files.

Reviewer #1: No

Reviewer #2: **Yes: **Bayan Al Omari

---

## [Author Response · Author response to Decision Letter 0]

17 Oct 2023

Author's Response to Reviewers' and Academic Editors' Comments: 

Manuscript number: PONE-D-23-23873

Title: Spatial Distribution of Home Delivery and Baby not Receive Postnatal Check-up within 2 Months after Birth in Ethiopia: Bivariate Analysis

We are thankful to the academic editor's and reviewers for the thoroughly review of our manuscript, encouraging remarks, their valuable comments and gave us the chance to resubmit the revised version. Following are replays that have been addressed in response to the academic editor's and reviewers' comments: 

 Academic Editors' Comments: 

Comment: 

Response: 

In order to comply with the journal's standards, the manuscript was thoroughly revised using the PLOS ONE’s style template, which can be seen in the links provided above (please see the marked-up and unmarked copies of the revised manuscript). 

Comment:

Response: 

Dear Editor, thank you for your feedback and guidance. We apologize for the confusion regarding the availability of our data. Upon reviewing the PLOS ONE data availability policy, we understand that PLOS only allows data to be available upon request if there are legal or ethical restrictions on sharing data publicly. We have updated the data availability statement accordingly and mentioned the changes in the revised cover letter

There is legal restrictions on sharing a de-identified data set as it is owned by a third-party (Demographic and health survey program). A written permission was obtained from the major demographic and health survey (DHS) program after online request to the program via the link www.dhsprogram.com. The permission letter was attached as file type “other” in the previous submission. Any interested body can register and access the data through the link provided above. 

Comment:

Response: 

Data from the major demographic and health survey program are accessible due to restrictions imposed by third parties. The institute can be contacted by interested researchers through online registration via the link www.dhsprogram.com to request access to the data. Following the editor's suggestions, the manuscript's data availability section has been revised and it was mentioned in the revised cover letter. 

Comment: 

4. We note that Figures 2, 3 and 4 in your submission contain map images which may be copyrighted. All PLOS content is published under the Creative Commons Attribution License (CC BY 4.0), which means that the manuscript, images, and Supporting Information files will be freely available online, and any third party is permitted to access, download, copy, distribute, and use these materials in any way, even commercially, with proper attribution. For these reasons, we cannot publish previously copyrighted maps or satellite images created using proprietary data, such as Google software (Google Maps, Street View, and Earth). For more information, see our copyright guidelines: http://journals.plos.org/plosone/s/licenses-and-copyright.

We require you to either present written permission from the copyright holder to publish these figures specifically under the CC BY 4.0 license, or (2) remove the figures from your submission:

a. you may seek permission from the original copyright holder of Figures 2, 3 and 4 to publish the content specifically under the CC BY 4.0 license. 

 Response: 

It is legitimate and crucial to address the problem of map copyright. Before utilizing it, we were aware of that. The maps depict the official map of Ethiopia and utilized for the purpose of illustrating some findings of the study. The maps were not copyrighted directly from third parties, rather drown by the current author using Arc Map version 10.8 software. The shape files we used to create the maps belonged to the Ethiopian Central Statistical Agency (CSA), later named as Ethiopian statistical service (ESS). We are thankful to the Ethiopian statistical service (ESS) for providing the shape files. 

Reviewers' comments:

Reviewer's Responses to Questions

Comments to the Author

1. Is the manuscript technically sound, and do the data support the conclusions?

Reviewer #1: Yes

Reviewer #2: Yes

Response: It is okay, thank you. 

2. Has the statistical analysis been performed appropriately and rigorously? 

Reviewer #1: Yes

Reviewer #2: I Don't Know

Response:

 We would like to aware the second reviewer that the statistical analysis has been performed appropriately and rigorously.

3. Have the authors made all data underlying the findings in their manuscript fully available?

Reviewer #1: Yes

Reviewer #2: Yes

Response: It is okay, thank you. 

4. Is the manuscript presented in an intelligible fashion and written in standard English?

Reviewer #1: Yes

Reviewer #2: No

Response: 

We have taken a step to correct typographical or grammatical errors that need to be corrected (see the marked-up copy of the manuscript in the revised submission). 

5. Review Comments to the Author

Reviewer #1: 

A good effort's to highlight this topic by providing with detailed statistical analysis, but there are many already provided literatures which focuses on postnatal checkup and delivery care. This study cannot add much new information to the existing literature.

Response: 

It is important for research studies to provide new insights or contribute novel findings to the field. Even though there are already numerous published studies that have extensively covered the topic of postnatal checkups and delivery care, the current manuscript should be assessed for its potential to add new information, perspectives, or approaches to the existing literatures. By this study it is tried to present a detailed statistical analysis, such as spatial and bivariate analysis of the two response variables (Delivery care and postnatal check-ups), thus it is essential to evaluate its potential contribution. 

Reviewer #2: Comments to Authors: 

Title: 

• The title of the study is relatively good. However, the authors should revise the title to be more inclusive and precise to the study’s objectives and aims (such as the determinants of PNC utilization).

Response: 

Dear reviewer, thank you for your guidance on revising the title of our study. We appreciate your suggestion and have made the necessary changes. The revised title now reads: “Spatial patterns and determinants of low utilization of delivery care service and postnatal check-up within 2 months following birth in Ethiopia: Bivariate analysis” (see the marked-up copy of the revised submission)

Introduction:

• It is clear, concise, and straightforward, and the rationale and aim of the study are well-stated. In addition, important definitions were illustrated clearly.

Response: It is okay, thank you.

Methods: 

• The authors conducted a database study and obtained ethical approval as requested.

• Sampling method was illustrated properly.

Response: It is okay, thank you. 

Results:

• The authors conducted a thorough analysis of the results, supported by sufficient amount of data justifying the conclusion. Nevertheless, the analysis assumed a linear relationship between dependent and independent variables, overlooking potential confounding factors that could impact the accuracy of correlations. Therefore, further research may be warranted to validate these hypotheses.

Response: 

Dear reviewer, thank you for your feedback on our analysis. We appreciate your point about potential confounding factors that could impact the accuracy of correlations and agree that further research may be warranted to validate our hypotheses. In future studies, we will take into consideration potential confounding factors and explore non-linear relationships between dependent and independent variables. This will help us provide a more comprehensive understanding of the determinants of delivery and postnatal care utilization. 

Discussion:

• In the discussion section, the authors have effectively compared their findings with those of similar studies and offered reasonable explanations for both the similarities and differences. It is crucial, however, to avoid over interpreting the data.

Response: 

Thank you for pointing out this potential issue and we appreciate your guidance in maintaining the integrity of our study. We understand the importance of not over-interpreting the data and have taken steps to avoid this in our revised submission (see the marked-up copy of the revised submission). 

• While the authors did highlight the clinical implications of their study's findings, it is worth noting that they missed addressing the study's limitations. It is advisable to have a transparent discussion of these limitations to provide a more comprehensive understanding of the research's scope and potential constraint.

Response: 

Thank you for your feedback. We have included a section in the revised submission dedicated to discussing the limitations of our study. This section provides an overview of potential constraints and acknowledges any biases or confounding factors that may have influenced our results. Additionally, we have included suggestions for future research to address these limitations and further explore the topic. These recommendations aim to guide other researchers in designing studies that can build upon our findings and contribute to the field (see the marked-up copy in the revised submission). 

• Further clinical application should be mentioned for policy makers, together with future direction and recommendation. 

Response: 

Thank you for your suggestion. We have added a section in the revised submission that discusses the potential clinical applications of our findings and provides recommendations for policy makers. This section highlights the practical implications of our study and how the results can be used to inform decision-making and policy development in the relevant field.

Furthermore, we have included a subsection on future directions and recommendations, which outlines specific areas for further research and investigation. These recommendations are intended to guide future studies and help expand our understanding of the topic (see the marked-up copy of the revised submission). 

Conclusion:

• Correctly answered the research question.

Response: thank you. 

• Authors should consider conciseness of conclusion to represent the whole findings.

Response: 

Thank you for your suggestion. We have revised the conclusion to ensure it represents the entirety of our findings in a concise manner. We have carefully summarized the key results and implications of our study, providing a clear and succinct conclusion that captures the main takeaways for both researchers and practitioners in the field (see the marked-up copy). 

Minor:

• English editing and proofreading by a native English speaker needed. The authors should correct some typos and language mistakes. 

Response: 

 We have taken the step of having the revised manuscript professionally edited and proofread for English language errors. However, we may still have some typos and language mistakes as none of the authors in this study are native English speakers. Thank you for your guidance and we apologize for any typos and language mistakes. 

Thank you dear editors and reviewers for your guidance and comments! 

Sincerely yours, 

Shegaw Mamaru Awoke

---

## [Decision Letter · Decision Letter 1]

27 Dec 2023

Spatial patterns and determinants of low utilization of delivery care service and postnatal check-up within 2 months following birth in Ethiopia: Bivariate analysis

PONE-D-23-23873R1

Dear Dr. Shegaw Amaru Awoke,

We’re pleased to inform you that your manuscript has been judged scientifically suitable for publication and will be formally accepted for publication once it meets all outstanding technical requirements.

Kind regards,

Gulzhanat Aimagambetova

Academic Editor

PLOS ONE

Reviewers' comments:

Reviewer's Responses to Questions

**Comments to the Author**

1. If the authors have adequately addressed your comments raised in a previous round of review and you feel that this manuscript is now acceptable for publication, you may indicate that here to bypass the “Comments to the Author” section, enter your conflict of interest statement in the “Confidential to Editor” section, and submit your "Accept" recommendation.

Reviewer #2: All comments have been addressed

Reviewer #3: All comments have been addressed

2. Is the manuscript technically sound, and do the data support the conclusions?

Reviewer #2: (No Response)

Reviewer #3: Yes

3. Has the statistical analysis been performed appropriately and rigorously? 

Reviewer #2: (No Response)

Reviewer #3: Yes

4. Have the authors made all data underlying the findings in their manuscript fully available?

Reviewer #2: (No Response)

Reviewer #3: Yes

5. Is the manuscript presented in an intelligible fashion and written in standard English?

Reviewer #2: (No Response)

Reviewer #3: Yes

6. Review Comments to the Author

Reviewer #2: (No Response)

Reviewer #3: Article entitled "Spatial patterns and determinants of low utilization of delivery care service and

postnatal check-up within 2 months following birth in Ethiopia: Bivariate analysis" addresses important issues of nonclinical childbirth practice and postnatal care. Authors address all previous comments. Authors use statistical methods to satisfactory level. Although, I do not agree with some terminology use, I think it is more of a question of preferences. Specifically, authors are recommended to revise the usage of terms "bivariate" and "binary", whether they appropriately use them in a context of the article. I do not have any further comments.

7. PLOS authors have the option to publish the peer review history of their article (what does this mean?). If published, this will include your full peer review and any attached files.

Reviewer #2: No

Reviewer #3: No

---

## [Editor Report · Acceptance letter]

15 Jan 2024

PONE-D-23-23873R1 

PLOS ONE

Dear Dr. Awoke, 

I'm pleased to inform you that your manuscript has been deemed suitable for publication in PLOS ONE. Congratulations! Your manuscript is now being handed over to our production team.

Kind regards, 

on behalf of

Dr. Gulzhanat Aimagambetova 

Academic Editor

PLOS ONE